# Prevalence of Congenital Heart Disease and Associated Factors among Infants with Myelomeningocele: A cross-sectional study

**Mohammed Nasir[1]\*, Bereket Tessema[1], Sura Markos[2]**

1 Department of Pediatrics and Child Health, Hawassa University, Hawassa, Ethiopia, 2 Department of Internal Medicine, Division of Cardiology, Hawassa University, Hawassa, Ethiopia

\* mn2572338@gmail.com

## Abstract

### Background

Meningomyelocele (MMC) is believed to commonly coexist with congenital heart disease (CHD), due to embryological pathways involving neural crest cells and risk factors such as folate and homocysteine metabolism abnormalities. However, evidence from developing countries is limited. Therefore, this study assessed the prevalence of CHD among infants with MMC and identified factors associated with the occurrence of CHD in infants with MMC.

### Methodology

This hospital-based cross-sectional study was conducted at Hawassa University Comprehensive Hospital between May 1 and August 1, 2025. Infants with MMC born from January 1, 2018, to January 1, 2024, who underwent echocardiographic evaluation were included. Data were collected through chart review. Prevalence of CHD in infants with MMC was described using frequencies and percentages, and binary logistic regression identified factors associated with CHD.

### Results

A total of 265 infants with MMC were included in this study. The prevalence of CHD among infants with MMC was 7.2%. The presence of maternal history of spontaneous abortion [AOR = 2.51, (95% CI: 1.12, 9.83); P = 0.03], maternal overweight or obesity [AOR = 2.93, (95% CI: 1.15, 7.98); P = 0.01], maternal diabetes mellitus [AOR = 2.22, (95% CI: 1.13, 8.45); P = 0.01], extracardiac anomalies in the infant [AOR = 2.94, (95% CI: 1.11, 8.78); P = 0.04] were associated with CHD in infants with MMC.

**Data availability statement:** All relevant data are within the manuscript and its Supporting Information files.

**Funding:** The author(s) received no specific funding for this work.

**Competing interests:** The authors have declared that no competing interests exist.

## Conclusion

CHD was relatively common among infants with MMC, affecting 7.2% of cases. Factors associated with CHD included maternal history of spontaneous abortion, maternal overweight or obesity, maternal diabetes mellitus, and the presence of extracardiac anomalies in the infant. These findings highlight the significance of routine cardiac evaluation and targeted maternal risk assessment in infants with MMC to improve early detection and management.

## Introduction

Meningomyelocele (MMC) is a common birth defect of the central nervous system that occurs when the neural tube fails to close during embryonic development, resulting from the combined influence of both genetic and environmental factors [1–3]. It is a type of open neural tube defect [1,2].

As of 2021, the highest burden of neural tube defects was observed in low Socio-Demographic Index regions, particularly in Eastern Sub-Saharan Africa. In 2020, Ethiopia had the highest burden of neural tube defects in Eastern Africa, with a prevalence of 0.32%, making it the second highest in Africa overall [4]. Myelomeningocele is the most common neural tube defect, with an incidence of approximately 4–5 per 10,000 pregnancies [5]. Of all neural tube defects, it accounts for approximately 36.4% to 51.4% of cases in Africa [4,6,7].

Neural crest cells play a critical role in the development of both the heart and neural tube [8]. Because of this common origin, congenital heart disease (CHD) is among the most commonly reported anomalies associated with neural tube defects, according to studies conducted in developed countries [9,10]. The association of CHD with neural tube defect has high clinical significance, as a study done in the USA showed infants who had both MMC and CHD had significantly longer hospital stays, higher total charges, and increased mortality compared to those with MMC or CHD alone [11]. Therefore, identifying CHD in children with MMC is essential in the integrated management of meningomyelocele and CHD [12].

Echocardiographic evaluation in infants with MMC is a critical diagnostic tool as it plays a vital role in the preoperative assessment for surgical repair [12]. Early diagnosis and management of CHD in patients with MMC is essential to prevent complications such as paradoxical embolism during shunt procedures and venous air embolism during cranial surgeries, particularly in the presence of intracardiac shunts, and infective endocarditis from frequent genitourinary manipulations [13]. The additional significance of identifying CHD in patients with MMC lies in anticipating, preventing, and enabling early management of cardiovascular complications, as a study from the United States reported that approximately 17.9% of 134 patients with MMC experienced cardiac events such as cardiac arrest, bradycardia, tachycardia, hypotension, and arrhythmias either intraoperatively or postoperatively [14]. Although identifying CHD in infants with neural tube defects is clinically important, data on the prevalence of CHD and factors associated with its occurrence among infants with

MMC in developing countries remain scarce. Therefore, this study aimed to assess the prevalence of CHD and the factors associated with its occurrence in infants with MMC.

## Methods

### Study area, period, and design

A hospital-based cross-sectional study was conducted at Hawassa University Comprehensive Specialized Hospital (HUCSH) in Hawassa, Ethiopia, from May 1 to August 1, 2025. The study included infants with MMC born between January 2018 and January 2024 who underwent echocardiographic evaluation.

HUCSH is the teaching hospital of Hawassa University's College of Health Sciences and School of Medicine, providing comprehensive care through multiple adult and pediatric units. Infants with myelomeningocele (MMC) are followed in the pediatric neurology unit, which delivers integrated, multidisciplinary care led by a pediatric neurologist and a neurosurgeon. Pediatricians conduct initial evaluation of infants with MMC in the neonatal or general pediatric outpatient units through detailed clinical examination and appropriate imaging studies. Following this assessment, patients are referred to the pediatric neurology unit for specialized follow-up and coordinated multidisciplinary management. Cardiac evaluations are routinely performed for all infants with MMC by two pediatric cardiologists. The assessment begins with a comprehensive physical examination, followed by trans-thoracic echocardiography using Sonoscape A6 and Mindray Zonare Z (Pro ZS3) systems. Infants diagnosed with CHD receive ongoing care and follow-up in the pediatric cardiology unit.

### Study population and sample size

The study population included all infants diagnosed with MMC in pediatric outpatient and inpatient units in HUCSH from 2018 to 2024. We calculated the sample size using a single population proportion formula, assuming a 50% prevalence of CHD among infants with MMC, as no similar research has been conducted in Ethiopian infants, with 5% precision and a 95% confidence level, yielding a sample size of 384. Since the total number of infants with MMC in our hospital was less than 10,000 (N = 265), we applied a finite population correction formula:

$$nf = (ni \times N)/ (ni + N)$$

Where nf is the final sample size, ni is the initial sample size, and N is the population size. Substituting the values gave:

$$nf = (384 \times 265)/(384 + 265) \approx 157$$

To account for potential non-respondents, 10% was added, resulting in a total sample size of 173. However, because the total number of infants with MMC attending follow-up in our hospital was small and manageable, we ultimately recruited all 265 infants with MMC (Fig 1).

### Eligibility criteria

All infants diagnosed with MMC based on physical examination and ultrasound who underwent echocardiographic evaluation for CHD were included in the study. Infants were excluded if their medical records were incomplete or if echocardiography had not been performed.

### Data collection tools and methods

Data were collected using a structured questionnaire specifically developed for this study, following a thorough literature review. The questionnaire was then pretested on infants with MMC who were not part of the main study at Adare Hospital

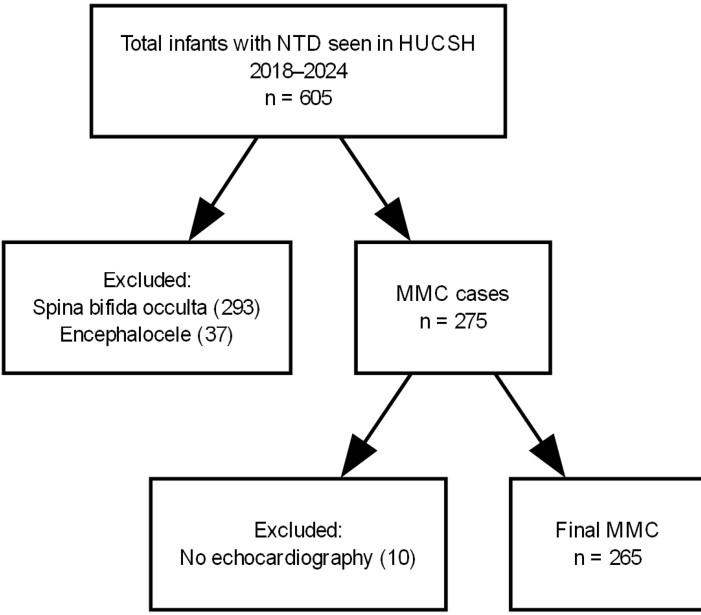

**Fig 1. Patient recruitment flow chart.**

in Hawassa (General Hospital in Hawassa). Data collection was carried out by pediatric residents who received two days of training on the data collection tools, the significance of the study, and ethical issues. After data collection, the quality of the collected data was checked daily by the investigators of this study.

### Variables

The dependent variable in this study was the presence of congenital heart disease (CHD) among infants with myelo-meningocele (MMC). Independent variables were selected based on clinical relevance, prior literature, and potential confounding effects. Maternal characteristics included age at delivery, residence, BMI-for-age category, parity, periconceptional folic acid supplementation, history of spontaneous abortion, presence of diabetes mellitus during pregnancy, drug and alcohol use during pregnancy, and previous history of a child with a neural tube defect. Neonatal characteristics included the infant's sex, the anatomical location of the MMC, and the presence and type of associated anomalies.

### Operational definitions

MMC was defined as a neural tube defect characterized by the protrusion of the spinal cord and meninges through a defect in the vertebral column, diagnosed by physical examination and confirmed with ultrasound in this study. CHD was defined as a structural abnormality of the heart present at birth, diagnosed by transthoracic echocardiography performed by a pediatric cardiologist. PFOs and small PDAs were not classified as CHDs and excluded from the CHD list in this study. In this study, CHD is categorized as severe (univentricular or biventricular) and non-severe (S1 File). The World Health Organization (WHO) Body Mass Index (BMI) classification was used to assess maternal nutritional status as underweight (≤ 18.5 kg/m²), normal weight (18.5–24.9 kg/m²), overweight (25.0–29.9 kg/m²), and obese (≥ 30.0 kg/m²) [15]. Similarly, the WHO criteria were used to diagnose diabetes mellitus in pregnancy [16]. Trans fontanelle ultrasound was used to diagnose hydrocephalus.

### Data analysis

Data completeness was manually verified before being entered into Microsoft Excel 2013 for data cleaning. The cleaned data were then exported to STATA version 17 for analysis. Categorical variables were summarized using frequencies and percentages, and the results were presented in tables. A binary logistic regression model was used to assess the association between the dependent variable and the independent variables. The bivariable analysis was first performed to examine the crude association between each independent variable and the outcome variable. Variables were selected for multivariable analysis based on a p-value <0.05 in the bivariable analysis, as well as clinical relevance, evidence from prior literature, and potential confounding effects. The strength of associations was reported using adjusted odds ratios (AORs) with 95% confidence intervals (CIs). A p-value less than 0.05 was considered statistically significant. Multicollinearity was assessed using the variance inflation factor (VIF), with a value greater than 10 indicating potential multicollinearity.

### Ethical approval and consent from participants

Ethical approval for this study was obtained from the Institutional Review Board of Hawassa University, College of Medicine and Health Sciences **(Reference No: IRB/375/16)**, in accordance with the principles of the Declaration of Helsinki and relevant local regulations. The study involved no direct contact with participants. Patient confidentiality was strictly maintained, and all data were fully anonymized before analysis. No personal identifiers were collected or disclosed.

## Results

### Sociodemographic, anthropometric, and obstetric characteristics of the mother

More than half (62.3%) of the mothers of infants with MMC were aged between 25 and 34 years at the time of delivery. The majority (66.8%) resided in rural areas. More than one-quarter (28.3%) of the mothers were classified as overweight or obese. Only a small proportion of mothers reported preconception folic acid intake (7.5%), alcohol consumption during pregnancy (4.5%), or drug use during pregnancy (5.7%). Nearly one-tenth (10.9%) had diabetes mellitus during pregnancy. Among these mothers, 18 (62.1%) had type 2 diabetes mellitus, 7 (24.1%) had type 1 diabetes mellitus, and 4 (13.8%) had gestational diabetes mellitus (Table 1).

### Sociodemographic and clinical characteristics of infants with MMC

Table 2 summarizes the sociodemographic and clinical characteristics of infants with MMC seen at Hawassa University Comprehensive Specialized Hospital from 2018 to 2024. After the exclusion of 10 infants with MMC who did not have echocardiographic evaluations, 265 infants with MMC were included in this study. The study population showed a slight male predominance, with a male-to-female ratio of 1.5. More than nine-tenths (90.9%) presented to the facility within the first three months of life. Prenatal diagnosis of MMC was made in only 13 infants, with congenital heart disease (CHD) also detected prenatally in three cases. More than half (66.8%) of the MMC cases were located in the lumbosacral region.

Associated neurological anomalies/complications were identified in 86(32.5%) of infants. Among these, 53(61.6%) infants had isolated hydrocephalus, while 19(22.1%) had hydrocephalus combined with paraplegia and 14(16.3%) hydrocephalus with meningitis.

Among 265 infants with MMC, 19 (7.2%) had CHD, of which 2 (10.5%) were severe (1 complete atrioventricular septal defect (AVSD) and 1 tetralogy of Fallot (TOF), and 17 (89.5%) were non-severe, including 10 secundum atrial septal defects (ASD), 4 perimembranous ventricular septal defects (PM VSD), and 3 patent ductus arteriosus (PDA). Only 3 out of 19 infants with MMC and CHD had an audible murmur and other abnormal sounds during physical examination. The types and frequency of CHDs are summarized in Fig 2. Extra cardiac, non-neurologic anomalies were identified in 45(17%) infants, and their distribution is summarized in Fig 3.

**Table 1. Sociodemographic, Anthropometric, and Obstetric Profiles of Mothers of Infants with MMC at Hawassa University Comprehensive Specialized Hospital, 2018–2024.**

| Variables | Category | Frequency | Percent |
|---|---|---|---|
| Age of the mother at delivery | 14-24 years | 35 | 13.2 |
| | 25-34years | 165 | 62.3 |
| | >34 years | 65 | 24.5 |
| Residence | Urban | 88 | 33.2 |
| | Rural | 177 | 66.8 |
| Anthropometry category based on BMI for age | Underweight | 38 | 14.3 |
| | Normal | 152 | 57.4 |
| | Overweight | 40 | 15.1 |
| | Obese | 35 | 13.2 |
| At least one ANC follow-up during pregnancy | No | 60 | 22.6 |
| | Yes | 205 | 77.4 |
| Periconceptional folic acid supplementation | No | 245 | 92.5 |
| | Yes | 20 | 7.5 |
| History of spontaneous abortion | No | 230 | 86.8 |
| | Yes | 35 | 13.2 |
| Maternal diabetes during pregnancy | No | 236 | 89.1 |
| | Yes | 29 | 10.9 |
| Maternal alcohol intake during pregnancy | No | 253 | 95.5 |
| | Yes | 12 | 4.5 |
| Maternal drug intake during pregnancy ¥ | No | 250 | 94.3 |
| | Yes | 15 | 5.7 |
| Previous history of a child with a neural tube defect | No | 252 | 95.1 |
| | Yes | 13 | 4.9 |

**Note:** ¥ drugs which were not prescribed by health care professionals.

## Factors associated with the co-occurrence of CHD in infants with MMC

From a multivariable analysis, a maternal history of spontaneous abortion was associated with 2.51 times higher odds of CHD [AOR = 2.51, (95% CI: 1.12, 9.83); P = 0.03]. The presence of maternal overweight or obesity increased the odds of CHD by 2.93 times [AOR = 2.93, (95% CI: 1.15, 7.98); P = 0.01], while maternal diabetes mellitus during pregnancy was associated with 2.22 times higher odds of CHD [AOR = 2.22, (95% CI: 1.13, 8.45); P = 0.01]. Additionally, the presence of extracardiac anomalies was associated with a 2.94-fold increase in the odds of CHD [AOR = 2.94, (95% CI: 1.11, 8.78); P = 0.04] (Table 3).

## Discussion

This study assessed the prevalence and patterns of CHD among infants with MMC at a referral center in Ethiopia, providing one of the few reports from the country, the African continent, and globally. Four factors associated with CHD in this population were identified. These findings provide important regional and continental insights to guide early detection, risk stratification, and the development of integrated management strategies for infants with MMC.

Disturbances in homocysteine and folate metabolism are thought to contribute to the development of both CHDs and NTDs, as folate plays a critical role in DNA synthesis, methylation, and cell proliferation during early embryogenesis [17].

**Table 2. Sociodemographic and clinical characteristics of infants with Myelomeningocele at Hawassa University Comprehensive Specialized Hospital, 2018–2024.**

| Variables | Category | Frequency | Percent |
|---|---|---|---|
| Age of the child at presentation | <28days | 122 | 46 |
| | 28days-3months | 119 | 44.9 |
| | 3months-6months | 24 | 9.1 |
| Sex | Male | 157 | 59.2 |
| | Female | 108 | 40.8 |
| MMC diagnosed prenatally | No | 252 | 95.1 |
| | Yes | 13 | 4.9 |
| Location of MMC | Cervicothoracic | 4 | 1.5 |
| | Thoracolumbar | 55 | 20.8 |
| | Lumbosacral | 177 | 66.8 |
| | Sacral | 29 | 10.9 |
| Associated neurological anomalies/complications | No | 179 | 67.5 |
| | Yes | 86 | 32.5 |
| Presence of CHD | No | 246 | 92.8 |
| | Yes | 19 | 7.2 |
| CHD diagnosed prenatally | No | 262 | 98.9 |
| | Yes | 3 | 1.1 |
| Presence of other extra-cardiac, non-neurologic anomalies | No | 220 | 83 |
| | Yes | 45 | 17 |

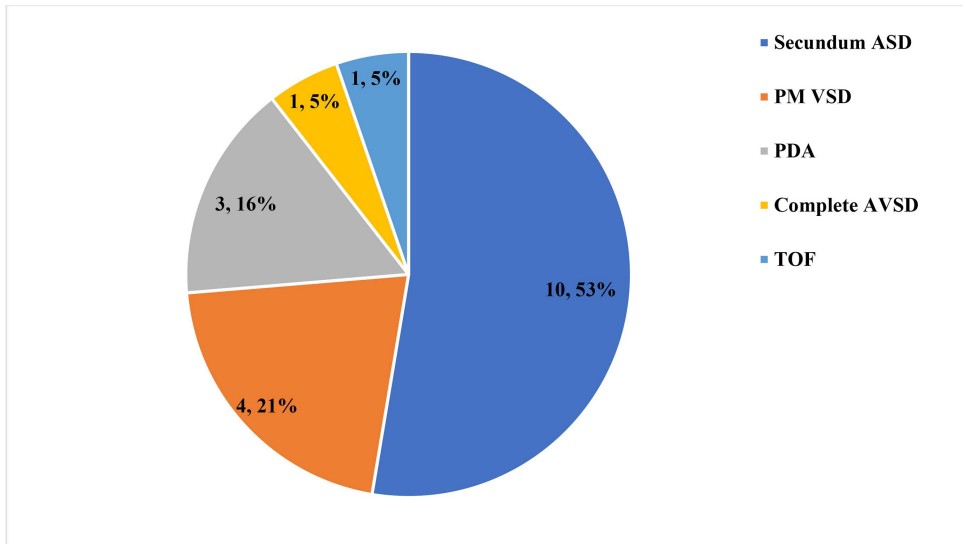

**Fig 2. Type and frequency of CHDs in infants with MMC at Hawassa University Comprehensive Specialized Hospital, 2018–2024.**

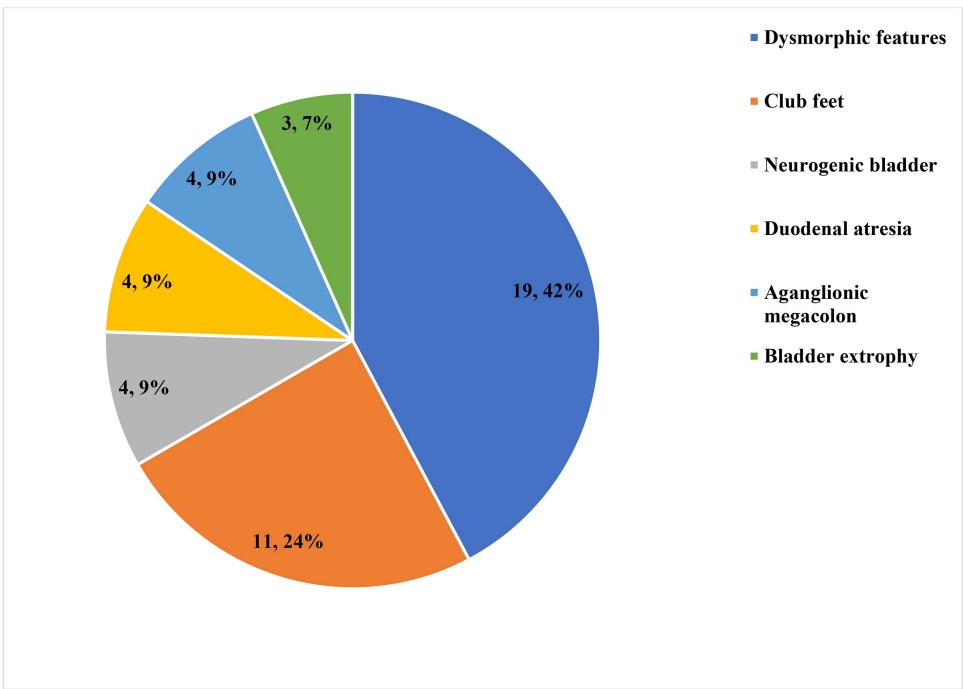

**Fig 3. Type and frequency of extra-cardiac non-neurologic anomalies in infants with MMC at Hawassa University Comprehensive Specialized Hospital, 2018–2024.**

**Table 3. Factors associated with the occurrence of CHD in infants with MMC at Hawassa University Comprehensive Specialized Hospital, 2018–2024.**

| Variable | Category | No CHD, n (%) | CHD, n (%) | COR (95% CI) | AOR (95% CI) |
|---|---|---|---|---|---|
| Maternal history of spontaneous abortion | No | 216 (87.8) | 13 (68.4) | 1 | 1 |
| | Yes | 30 (12.2) | 6 (31.6) | 3.32 (1.17–9.40) | 2.51 (1.12–9.83) |
| Maternal anthropometry (BMI category) | Underweight/Normal | 181 (73.6) | 9 (47.4) | 1 | 1 |
| | Overweight/Obese | 65 (26.4) | 10 (52.6) | 3.09 (1.20–7.95) | 2.93 (1.15–7.98) |
| Maternal diabetes mellitus during pregnancy | No | 222 (90.2) | 14 (73.7) | 1 | 1 |
| | Yes | 24 (9.8) | 5 (26.3) | 3.30 (1.09–9.97) | 2.22 (1.13–8.45) |
| Sex of the child | Male | 151 (61.4) | 6 (31.6) | 1 | 1 |
| | Female | 95 (38.6) | 13 (68.4) | 3.44 (1.26–9.37) | 3.26 (0.99–9.87) |
| Location of meningomyelocele | Cervicothoracic/Thoracolumbar | 50 (20.3) | 9 (47.4) | 3.53 (1.36–9.15) | 3.34 (0.82–9.33) |
| | Lumbosacral/Sacral | 196 (79.7) | 10 (52.6) | 1 | 1 |
| Presence of extracardiac anomaly | No | 208 (84.6) | 12 (63.2) | 1 | 1 |
| | Yes | 38 (15.4) | 7 (36.8) | 3.19 (1.18–8.63) | 2.94 (1.11–8.78) |

*Footnote:* **COR**: Crude odds ratio; **AOR**: Adjusted odds ratio; **CI**: Confidence interval. The reference category is indicated as **1**. Variables included in the multivariable logistic regression model were selected based on statistical criteria (p-value <0.05 in bivariable analysis) and clinical relevance, supported by evidence from prior literature and consideration of potential confounding effects. Maternal age was excluded from the multivariable model due to multicollinearity with a history of spontaneous abortion (Variance Inflation Factor = 11.3). Statistical significance was considered at $p < 0.05$.

This is supported by evidence showing a significant reduction in the risk of both conditions with periconceptional folate supplementation [18]; however, in our setting, few mothers received folate during the periconceptional period, which might have contributed to the high prevalence of CHD in infants with MMC in our cases.

In this study, the 7.2% prevalence of CHD in infants with MMC lies in the reported range of 2.7% to 37% in studies conducted across both developed and developing countries [11,13,19,20]. However, the wide variation in prevalence of CHD in infants with MMC across studies may be explained by differences in age groups, sex distribution, inclusion criteria for CHDs, periconceptional folate supplementation, and prevalence of risk factors.

The CHD spectrum in this cohort aligns with reported patterns from studies conducted in the same area in other regions. For example, in line with studies from the USA, Turkey, and Bangladesh, this study identified non-severe CHDs, such as atrial septal defects (ASDs) and ventricular septal defects (VSDs), as the most common congenital heart anomalies in infants with MMC [13,21,22]. Additionally, this study demonstrated that severe critical congenital heart diseases were uncommon. Consistent with this finding, a study from the United States reported Tetralogy of Fallot as a rare association [13]. Another U.S. study, similar to ours, also identified atrioventricular septal defect (AVSD) as a rare co-occurrence [23].

Surprisingly, most infants with meningomyelocele and associated CHD did not show abnormal cardiac auscultation findings, as seen in this study and a study from the USA [13]. Therefore, we relied mainly on echocardiography for the diagnosis of CHDs.

In this study, a maternal history of spontaneous abortion was one of the associated factors that increased the odds of the occurrence of CHD in infants with MMC. This association may reflect shared underlying factors, such as genetic mutations, chromosomal abnormalities, unidentified maternal autoimmune disorders, thrombophilia, or metabolic diseases, that contribute to both spontaneous abortion and the development of structural anomalies, including neural tube defects and CHDs [24,25].

Maternal overweight/obesity and diabetes have been shown to increase the odds of CHD in infants with MMC in this study, which was in line with numerous other studies [26–30]. For instance, a Finnish study reported that maternal diabetes increases the risk of nearly all types of CHDs. Additionally, it showed the association of maternal overweight with left ventricular outflow tract obstruction and ventricular septal defects, while maternal obesity was associated with more complex cardiac anomalies and right ventricular outflow tract obstruction in the newborn [31]. The association of maternal overweight, obesity, and diabetes can be attributed to the teratogenic effect of high maternal hyperglycemia in the fetus after crossing the placenta [32]. Hyperinsulinemia, high inflammation, and oxidative stress are also postulated to have a contribution [33,34].

The absence of a clear association between maternal drug use, alcohol consumption, or inadequate folate supplementation during early pregnancy and CHD in infants with MMC may not reflect the true relationship and could be due to the small sample sizes in this study and previous studies [13,21,22]. Larger studies are needed to accurately evaluate these potential associations.

The presence of extracardiac anomalies increased the odds of occurrence of CHD in infants with MMC in this study, which might be explained by the fact that both NTDs and CHD are components of various syndromic and association patterns, such as Edward's syndrome, Patau's syndrome, and VACTERL associations [35].

## Conclusion

CHD was relatively common among infants with MMC, affecting 7.2% of cases. The maternal history of spontaneous abortion, maternal overweight or obesity, diabetes mellitus in the mother, and the presence of extracardiac anomalies in the infant were significant factors associated with the occurrence of CHD. These findings highlight the need for routine cardiac evaluation and targeted maternal risk assessment in infants with MMC to improve early detection and management.

## Limitations of the study

This study has several limitations. First, its retrospective design may introduce bias in the classification of both outcome and independent variables. Second, there is potential survival bias, as 10 newborns who were not screened for CHD at birth were lost to follow-up and never underwent echocardiography, leaving their cardiac status unknown. Third, only ultrasound was used to assess brain anatomy and the spinal cord, rather than MRI, which provides more detailed structural information. Finally, the multivariable logistic regression model included six predictor variables despite the relatively small number of CHD events (n = 19). This limited number of events per variable may have reduced the stability of the estimates and resulted in relatively wide confidence intervals. While we used standard logistic regression methods in this study, we acknowledge that alternative approaches such as penalised regression methods (e.g., Firth logistic regression) are specifically designed to provide more robust and less biased estimates in settings with sparse outcome data and may be preferable in future studies addressing similar research questions. Despite these limitations, this study provides valuable benchmark data on the prevalence of CHD in infants with MMC and identifies independent maternal and fetal factors associated with CHD. These findings can inform early screening, risk stratification, and comprehensive intervention strategies.

## Supporting information

**S1 File. CHD types categorized by severity (XLSX).** Classification of congenital heart disease types according to severity categories used in the study.
(XLSX)

**S2 File. Minimal dataset (XLSX).** Anonymized dataset used for the analysis.
(XLSX)

## Acknowledgments

We sincerely thank the staff of Hawassa University Comprehensive Specialized Hospital for their support during data collection, and we also acknowledge all data collectors for their dedication and assistance.

## Author contributions

**Conceptualization:** Mohammed Nasir, Bereket Tessema, Sura Markos.

**Data curation:** Mohammed Nasir, Bereket Tessema, Sura Markos.

**Formal analysis:** Mohammed Nasir, Bereket Tessema.

**Funding acquisition:** Mohammed Nasir, Bereket Tessema.

**Investigation:** Mohammed Nasir, Bereket Tessema.

**Methodology:** Mohammed Nasir, Bereket Tessema.

**Project administration:** Mohammed Nasir, Bereket Tessema.

**Resources:** Mohammed Nasir, Bereket Tessema.

**Software:** Mohammed Nasir, Bereket Tessema.

**Supervision:** Mohammed Nasir, Bereket Tessema.

**Validation:** Mohammed Nasir, Bereket Tessema.

**Visualization:** Mohammed Nasir, Bereket Tessema.

**Writing – original draft:** Mohammed Nasir, Bereket Tessema.

**Writing – review & editing:** Mohammed Nasir, Bereket Tessema.

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
