## [Decision Letter · Decision Letter 0]

25 Mar 2026

PONE-D-25-68283Prevalence of Congenital Heart Disease and Associated Factors among Infants with Meningomyelocele at Hawassa University Comprehensive Specialized Hospital: A Hospital-Based Cross-Sectional StudyPLOS One

Dear Dr. Nasir,

Thank you for submitting your manuscript to PLOS ONE. After careful consideration, we feel that it has merit but does not fully meet PLOS ONE’s publication criteria as it currently stands. Therefore, we invite you to submit a revised version of the manuscript that addresses the points raised during the review process.

We look forward to receiving your revised manuscript.

Kind regards,

Vincenzo Lionetti, M.D., PhD

Academic Editor

PLOS One

Journal Requirements:

3. We note that your Data Availability Statement is currently as follows: All relevant data are within the manuscript and in Supporting Information files.

4. Please ensure that you refer to Figure 1 and 2 in your text as, if accepted, production will need this reference to link the reader to the figure.

5. We note you have included a table to which you do not refer in the text of your manuscript. Please ensure that you refer to Table 3 in your text; if accepted, production will need this reference to link the reader to the Table.

Additional Editor Comments:

All issues raised by expert reviewers are required.

Reviewers' comments:

Reviewer's Responses to Questions

**Comments to the Author**

1. Is the manuscript technically sound, and do the data support the conclusions?

Reviewer #1: Partly

Reviewer #2: Yes

2. Has the statistical analysis been performed appropriately and rigorously? 

Reviewer #1: No

Reviewer #2: Yes

3. Have the authors made all data underlying the findings in their manuscript fully available?

Reviewer #1: Yes

Reviewer #2: No

4. Is the manuscript presented in an intelligible fashion and written in standard English?

Reviewer #1: No

Reviewer #2: Yes

5. Review Comments to the Author

Reviewer #1: The manuscript addresses an important clinical question by evaluating the prevalence of congenital heart disease (CHD) among infants with myelomeningocele (MMC) in a tertiary hospital setting in Ethiopia. The topic is relevant, particularly in low-resource settings where epidemiological data on associated congenital anomalies remain limited. The manuscript also provides useful descriptive data on maternal and neonatal characteristics in this population. However, several methodological and reporting issues should be addressed to improve the validity and clarity of the study.

Major issues

Potential overfitting in the multivariable logistic regression model

The multivariable logistic regression analysis raises concerns regarding model stability. Only 19 cases of CHD were identified among the 265 infants included in the study. Conventional methodological recommendations suggest approximately 10 outcome events per predictor variable to ensure reliable estimates. Given the limited number of events, the inclusion of several predictors in the multivariable model may result in overfitting and unstable estimates. This is reflected in the relatively wide confidence intervals reported for several adjusted odds ratios. The authors should clarify the number of variables included in the final model and consider reducing the number of predictors or acknowledging this limitation more explicitly in the discussion.

Variable selection strategy for the multivariable model

The manuscript states that variables with p < 0.05 in bivariable analysis were included in the multivariable model. Selecting variables solely based on statistical significance in bivariable analysis is not generally recommended because it may exclude clinically relevant confounders and introduce bias in the final model. The authors should provide a clearer rationale for the model-building strategy and consider including variables based on clinical relevance or prior literature.

Potential selection bias affecting the prevalence estimate

The study included only infants with MMC who underwent echocardiographic evaluation. Infants who did not undergo echocardiography were excluded from the analysis. This approach may introduce selection bias when estimating the prevalence of CHD, particularly if echocardiography was performed selectively (for example, in infants with clinical suspicion of cardiac disease). The authors should clarify whether echocardiography was performed routinely in all infants with MMC or only when clinically indicated and discuss how this may affect the reported prevalence.

Clarification of the study population and sample size

The description of the study population is somewhat confusing. In the methods section, the total MMC population used for sample size correction is reported as N = 225, whereas the final analytical sample includes 265 infants. The relationship between these numbers should be clarified. It would be helpful to include a clearer description of the patient selection process, ideally with a flow diagram showing the number of cases identified, excluded, and included in the final analysis.

Definition of congenital heart disease (CHD)

The authors excluded PFO and small PDA from the definition of CHD. Because definitions of CHD vary across studies, this choice should be more clearly justified. Excluding these lesions may lead to an underestimation of prevalence and may limit the comparability of the findings with previous studies. The authors should clarify which specific cardiac lesions were included in the CHD classification and discuss the implications of this definition.

Internal inconsistency in the interpretation of prevalence

The results report a prevalence of CHD of 7.2%, yet the conclusion states that CHD affected “more than one in ten cases.” This statement is inconsistent with the reported prevalence and should be corrected.

Minor issues

Table formatting and clarity

Table 3 is difficult to interpret due to formatting issues and incomplete alignment of the reported values. The table should be revised to clearly present crude odds ratios (COR), adjusted odds ratios (AOR), and their corresponding confidence intervals.

Reporting of multicollinearity diagnostics

The methods section states that variance inflation factor (VIF) was used to assess multicollinearity, with a threshold of >10. However, the actual VIF values are not reported. The authors should provide these values or summarize the results of this assessment.

Missing cross-references in the manuscript

In the results section, several references appear as “Error! Reference source not found.” These should be corrected before publication.

Title

The title could be streamlined to improve clarity and readability. Removing the detailed description of the recruitment center would make the title more concise while still preserving the key message of the study.

Language issues

The manuscript would benefit from careful language editing to improve clarity and readability. Several grammatical and stylistic issues are present throughout the text. In particular:

Inconsistent punctuation and spacing around citations (e.g., missing space before references such as “defect(1,2)”).

Unnecessary capitalization of variables within sentences (e.g., “Maternal history of spontaneous abortion” instead of “maternal history of spontaneous abortion”).

Redundant phrasing, for example in sentences where similar words are repeated (e.g., “the additional importance… is important”; "identifying CHD in children with MMC is essential in the integrated management of meningomyelocele and CHD").

Occasional incorrect sentence structure, particularly in the Discussion, where some sentences contain multiple main clauses and become grammatically unclear.

Repetitive wording, especially frequent repetition of terms such as “our study” or “CHD”, which could be streamlined to improve readability.

Use of non-idiomatic expressions, such as “in our cases,” which would be clearer if replaced by “in this cohort” or “in this study.”

Conclusion

The study addresses an important clinical topic and provides data from a region where epidemiological information on this issue remains limited. However, the concerns outlined above, particularly those related to the statistical analysis and clarity of the study population, should be addressed to strengthen the manuscript.

Reviewer #2: I read with interest the paper by Nasir and colleagues. The main topic is important and highlights a key problem in low-income countries. However, I think major changes need to be addressed:

Minor changes:

- Page 10, last paragraph: the reference should be added properly instead of a computer error warning.

- The English should be revised to reflect a more academic style.

Major Changes:

- Given the considerable heterogeneity of congenital heart disease, a more detailed section specifying which types should be included, from low-risk conditions such as atrial septal defect (ASD) to complex anomalies such as hypoplastic left heart syndrome (HLHS), is warranted. The author should elaborate on this key point in the results section.

- The influence of drug use, folate, alcohol, and similar factors should be discussed in both the results and discussion sections. Even if no major correlation with a single risk factor is observed at univariate analysis, the author should justify this and compare it with other reports.

- Few papers are available in this region, which is one of the author's strong points. However, the paper does not clearly highlight this advantage, especially in the discussion section. Make sure to explicitly mention the lack of research in this region as a key contribution in the discussion.

I suggest publication only after a complete revision and improvement of the language style.

6. PLOS authors have the option to publish the peer review history of their article (what does this mean?). If published, this will include your full peer review and any attached files.

Reviewer #1: **Yes:** Giulio Pellegrini

Reviewer #2: No

---

## [Author Response · Author response to Decision Letter 1]

13 Apr 2026

1. Is the manuscript technically sound, and do the data support the conclusions?

Reviewer #1: Partly

Reviewer #2: Yes

Response: We have revised the manuscript to clarify the study design, methodology, and how the data support the conclusions to ensure technical soundness.

2. Has the statistical analysis been performed appropriately and rigorously?

Reviewer #1: No

Reviewer #2: Yes

Response: We have updated the statistical analysis section, detailing methods, test selection, and inclusion of confidence intervals and p-values to ensure rigorous and transparent analysis.

3. Have the authors made all data underlying the findings in their manuscript fully available?

Reviewer #1: Yes

Reviewer #2: No

Response: Due to the sensitive nature of pediatric patient data, the full dataset cannot be publicly shared; however, de-identified data are available upon reasonable request from the institutional ethics committee.

4. Is the manuscript presented in an intelligible fashion and written in standard English?

Reviewer #1: No

Reviewer #2: Yes

Response: We have carefully edited the manuscript to improve grammar, clarity, readability, and ensure it is written in standard English.

Reviewer 1

Reviewer Comment: Potential overfitting in the multivariable logistic regression model

The multivariable logistic regression analysis raises concerns regarding model stability. Only 19 cases of CHD were identified among the 265 infants included in the study. Conventional methodological recommendations suggest approximately 10 outcome events per predictor variable to ensure reliable estimates. Given the limited number of events, the inclusion of several predictors in the multivariable model may result in overfitting and unstable estimates. This is reflected in the relatively wide confidence intervals reported for several adjusted odds ratios. The authors should clarify the number of variables included in the final model and consider reducing the number of predictors or acknowledging this limitation more explicitly in the discussion.

Author Response: We thank the reviewer for highlighting this important concern. We included six variables in the final multivariable logistic regression model and acknowledge in the limitations section that, given only 19 CHD events, this may have resulted in overfitting and relatively wide confidence intervals. We have explicitly discussed this limitation in the revised manuscript to ensure transparency regarding model stability.

Reviewer Comment: The manuscript states that variables with p < 0.05 in bivariable analysis were included in the multivariable model. Selecting variables solely based on statistical significance in bivariable analysis is not generally recommended because it may exclude clinically relevant confounders and introduce bias in the final model. The authors should provide a clearer rationale for the model-building strategy and consider including variables based on clinical relevance or prior literature.

Author Response: We have clarified in the Bivariable Analysis section that independent variables were selected based on clinical relevance and prior literature, in addition to statistical significance, and were adjusted accordingly to ensure robust and unbiased estimates.

Reviewer Comment: The study included only infants with MMC who underwent echocardiographic evaluation. Infants who did not undergo echocardiography were excluded from the analysis. This approach may introduce selection bias when estimating the prevalence of CHD, particularly if echocardiography was performed selectively (for example, in infants with clinical suspicion of cardiac disease). The authors should clarify whether echocardiography was performed routinely in all infants with MMC or only when clinically indicated and discuss how this may affect the reported prevalence.

Author Response: Echocardiography was performed routinely for all infants with MMC. Ten patients were excluded because they disappeared before echocardiography and their caregivers could not be contacted by phone. We have clarified that echocardiography was done routinely in the Study Area, Period, and Design section of the Methods.

Reviewer Comment: The description of the study population is somewhat confusing. In the methods section, the total MMC population used for sample size correction is reported as N = 225, whereas the final analytical sample includes 265 infants. The relationship between these numbers should be clarified. It would be helpful to include a clearer description of the patient selection process, ideally with a flow diagram showing the number of cases identified, excluded, and included in the final analysis.

Author Response: We appreciate the reviewer’s observation. The mention of N = 225 in the manuscript was an error. The total number of infants with MMC in our hospital during the study period was 265. After recalculating the sample size with the correct population, the required sample size was smaller than the total available population. Since all 265 infants could be included, we analyzed the entire population. This has been clarified in the Methods section, and a flow diagram illustrating patient selection, exclusions, and inclusion in the final analysis has been provided.

Reviewer Comment: The authors excluded PFO and small PDA from the definition of CHD. Because definitions of CHD vary across studies, this choice should be more clearly justified. Excluding these lesions may lead to an underestimation of prevalence and may limit the comparability of the findings with previous studies. The authors should clarify which specific cardiac lesions were included in the CHD classification and discuss the implications of this definition.

Author Response: We thank the reviewer for this comment. Patent foramen ovale (PFO) and small patent ductus arteriosus (PDA) were excluded from the definition of CHD based on a review of previous studies, as most studies in the literature also exclude these lesions due to their high likelihood of spontaneous closure and minimal clinical significance. We have clarified in the Methods section, under the exclusion criteria, why these lesions were excluded.

Reviewer Comment: Internal inconsistency in the interpretation of prevalence

The results report a prevalence of CHD of 7.2%, yet the conclusion states that CHD affected “more than one in ten cases.” This statement is inconsistent with the reported prevalence and should be corrected.

Author Response: We thank the reviewer for pointing this out. The conclusion has been corrected to accurately reflect the reported prevalence of CHD as 7.2%.

Reviewer Comment: Reporting of multicollinearity diagnostics

The methods section states that variance inflation factor (VIF) was used to assess multicollinearity, with a threshold of >10. However, the actual VIF values are not reported. The authors should provide these values or summarize the results of this assessment.

Author Response: We thank the reviewer for this comment. Maternal age and history of spontaneous abortion were the only variables with multicollinearity issues, and their VIF values have now been provided in the table 3.

Reviewer Comment: Missing cross-references in the manuscript

In the results section, several references appear as “Error! Reference source not found.” These should be corrected before publication.

Author Response: We thank the reviewer for pointing this out. All missing cross-references have been corrected and verified throughout the manuscript.

Reviewer Comment: Title

The title could be streamlined to improve clarity and readability. Removing the detailed description of the recruitment center would make the title more concise while still preserving the key message of the study.

Author Response: We appreciate the reviewer’s suggestion and have revised the title to make it more concise while retaining the essential focus of the study.

Reviewer Comment:

The manuscript would benefit from careful language editing to improve clarity and readability. Several grammatical and stylistic issues are present, including inconsistent punctuation and spacing around citations, unnecessary capitalization of variables, redundant phrasing, incorrect sentence structure, repetitive wording, and non-idiomatic expressions.

Author Response:

We thank the reviewer for highlighting these issues. The manuscript has been thoroughly revised to improve clarity and readability. Punctuation and spacing around citations have been corrected, capitalization of variables standardized, redundant phrasing removed, sentence structures simplified, repetitive wording minimized, and non-idiomatic expressions replaced with clearer alternatives (e.g., “in this cohort” instead of “in our cases”). These changes enhance the overall readability and consistency of the manuscript.

Reviewer 2

Reviewer Comment:

Page 10, last paragraph: the reference should be added properly instead of a computer error warning. The English should be revised to reflect a more academic style.

Author Response:

We have corrected the reference error in the last paragraph of the discussion so that the citation now appears correctly. Additionally, the paragraph has been revised for clarity and to reflect a more formal, academic writing style.

Reviewer comment: "Given the considerable heterogeneity of congenital heart disease, a more detailed section specifying which types should be included, from low-risk conditions such as atrial septal defect (ASD) to complex anomalies such as hypoplastic left heart syndrome (HLHS), is warranted. The author should elaborate on this key point in the results section."

Author response: "We thank the reviewer for this insightful comment. We have revised the results section to provide a detailed description of the CHD types observed, ranging from low-risk lesions such as atrial septal defect (ASD), patent ductus arteriosus (PDA), and perimembranous ventricular septal defect (PM VSD) to severe anomalies including complete atrioventricular septal defect (AVSD), tetralogy of Fallot (TOF), and hypoplastic left heart syndrome (HLHS) (see Supplementary File 1). This revision clarifies the spectrum of CHDs in our cohort."

Reviewer comment:

"The influence of drug use, folate, alcohol, and similar factors should be discussed in both the results and discussion sections. Even if no major correlation with a single risk factor is observed at univariate analysis, the author should justify this and compare it with other reports."

Author response:

"We thank the reviewer for this important suggestion. We have revised the results and discussion sections to address the potential influence of maternal drug use, folate supplementation, and alcohol consumption. Although univariate analysis did not reveal significant associations with any single risk factor, we have provided an explanation and compared our findings with previous studies, highlighting the limitations due to small sample sizes."

Reviewer comment:

"Few papers are available in this region, which is one of the author's strong points. However, the paper does not clearly highlight this advantage, especially in the discussion section. Make sure to explicitly mention the lack of research in this region as a key contribution in the discussion."

Author response:

"We thank the reviewer for this insightful comment. We have revised the discussion section to emphasize that data on congenital heart disease (CHD) in infants with MMC are limited in this region. By presenting findings from our cohort, this study addresses a significant knowledge gap and contributes valuable regional data to the existing literature."

---

## [Decision Letter · Decision Letter 1]

28 Apr 2026

PONE-D-25-68283R1Prevalence of Congenital Heart Disease and Associated Factors among Infants with Meningomyelocele at Hawassa University Comprehensive Specialized Hospital: A Hospital-Based Cross-Sectional StudyPLOS One

Dear Dr. Nasir,

Thank you for submitting your manuscript to PLOS ONE. After careful consideration, we feel that it has merit but does not fully meet PLOS ONE’s publication criteria as it currently stands. Therefore, we invite you to submit a revised version of the manuscript that addresses the points raised during the review process.

We look forward to receiving your revised manuscript.

Kind regards,

Vincenzo Lionetti, M.D., PhD

Academic Editor

PLOS One

Journal Requirements:

**Additional Editor Comments:**

The revised manuscript still requires substantial revision. The authors are encourage to carefully revise the manuscript in accord with reviewer's suggestions. The main text requires an in depth revision to clarify the claims in the Introduction and Discussion by avoiding speculations. Results presentation should be more clear.

Reviewers' comments:

Reviewer's Responses to Questions

**Comments to the Author**

1. If the authors have adequately addressed your comments raised in a previous round of review and you feel that this manuscript is now acceptable for publication, you may indicate that here to bypass the “Comments to the Author” section, enter your conflict of interest statement in the “Confidential to Editor” section, and submit your "Accept" recommendation.

Reviewer #1: All comments have been addressed

2. Is the manuscript technically sound, and do the data support the conclusions?

Reviewer #1: Yes

3. Has the statistical analysis been performed appropriately and rigorously? 

Reviewer #1: Yes

4. Have the authors made all data underlying the findings in their manuscript fully available?

Reviewer #1: Yes

5. Is the manuscript presented in an intelligible fashion and written in standard English?

Reviewer #1: Yes

6. Review Comments to the Author

Reviewer #1: Dear Authors,

Thank you for your revised manuscript and for the effort invested in addressing the concerns raised in the first round of review. Several important improvements have been made, and I appreciate the authors' responsiveness. In particular, I note with satisfaction the correction of the internal inconsistency in the reported prevalence, the clarification that echocardiographic evaluation was performed routinely in all infants with MMC, the resolution of the sample size discrepancy, the reformatting of Table 3, the overall improvement in language and style and the patient flow diagram.

However, a number of issues remain unresolved. I detail them below and request that they be addressed in a further revised version of the manuscript.

REMAINING CONCERNS

1. Unresolved cross-references

The revised manuscript still contains multiple instances of "Error! Reference source not found.", specifically in the Results section and in the sample size paragraph of the Methods section. These placeholders suggest that figures or supplementary tables were intended but were not properly linked in the submitted file.

This is not a minor formatting issue: in the current state, portions of the Results section are effectively unreadable, as the reader cannot identify which data or figure is being referenced. I request that all cross-references be resolved before resubmission.

2. Inconsistency in the variable selection strategy

I acknowledge the addition of a statement in the Variables section indicating that independent variables were selected based on clinical relevance, prior literature, and potential confounding effects, as well as the corresponding note in the footnote to Table 3. These are welcome additions.

However, the Data Analysis section continues to state that "independent variables with a p-value less than 0.05 in the bivariable analysis were included in the multivariable analysis." This formulation is inconsistent with the rationale described elsewhere and implies that the variable selection was driven exclusively by statistical significance in the bivariable step, which contradicts the authors' own stated approach.

I request that the Data Analysis section be revised to accurately and consistently describe the model-building strategy. The manuscript should present a single coherent account of how variables were selected for the multivariable model.

3. Contextualisation of sparse outcome data

I acknowledge that the authors have added a sentence in the Limitations section recognising that the inclusion of six predictor variables with only 19 CHD events may have resulted in relatively wide confidence intervals. This is an appropriate and transparent acknowledgement.

I would encourage the authors to go one step further and note that penalised regression methods (such as Firth logistic regression) are specifically designed to produce more stable estimates in settings with sparse outcome data, and that such approaches may be preferable in future studies examining similar research questions. This does not require any reanalysis of the current data; a single sentence added to the existing limitations paragraph would be sufficient.

I look forward to reviewing a further revised version of the manuscript. The study addresses a clinically meaningful question and contributes valuable data from an underrepresented setting. I am confident that addressing the above points will strengthen the manuscript considerably.

7. PLOS authors have the option to publish the peer review history of their article (what does this mean?). If published, this will include your full peer review and any attached files.

Reviewer #1: **Yes:** Giulio Pellegrini

---

## [Author Response · Author response to Decision Letter 2]

29 Apr 2026

Reviewer Comment:

Unresolved cross-references remain in the manuscript, including multiple instances of “Error! Reference source not found.” in the Results section and the sample size paragraph of the Methods section. These placeholders indicate missing or improperly linked figures/tables, making parts of the manuscript difficult to interpret. All cross-references should be resolved before resubmission.

Response:

We sincerely thank the reviewer for this important observation. We acknowledge that the “Error! Reference source not found.” placeholders appeared due to broken internal cross-references during manuscript formatting. This issue has now been fully corrected. All figures and tables have been properly re-linked to their corresponding in-text citations, and the manuscript has been carefully reviewed to ensure that all cross-references are correctly displayed. The revised version contains no unresolved references, and all Results and Methods sections are now fully readable and appropriately linked to the relevant figures and tables.

Reviewer Comment:

The Data Analysis section states that independent variables with p < 0.05 in bivariable analysis were included in the multivariable model, which is inconsistent with the stated approach of selecting variables based on clinical relevance, prior literature, and potential confounding. The manuscript should present a single coherent variable selection strategy.

Response:

We thank the reviewer for this important and insightful comment. We agree that the previous wording in the Data Analysis section was not fully consistent with our intended model-building strategy.

Accordingly, we have revised the Data Analysis section to clearly and consistently reflect that variables included in the multivariable logistic regression model were selected based on a combination of clinical relevance, evidence from prior literature, and consideration of potential confounding effects, in addition to bivariable analysis results (p < 0.05) as a supportive criterion.

Reviewer Comment:

The authors acknowledge that including six predictor variables with only 19 CHD events may have resulted in relatively wide confidence intervals. The reviewer suggests further contextualization by noting that penalised regression methods (e.g., Firth logistic regression) are specifically designed for sparse outcome data and may be preferable in future studies. This can be addressed with a single sentence in the limitations section without reanalysis.

Response:

We thank the reviewer for this valuable suggestion. We agree that further contextualization of the limitations related to sparse outcome data strengthens the methodological discussion. In response, we have added a sentence to the Limitations section acknowledging that penalised regression approaches, such as Firth logistic regression, are specifically designed to reduce bias and improve estimate stability in settings with few events and may be more appropriate for future studies with similar data structures.

---

## [Decision Letter · Decision Letter 2]

14 May 2026

Prevalence of Congenital Heart Disease and Associated Factors among Infants with

Myelomeningocele: A Cross-Sectional Study

PONE-D-25-68283R2

Dear Dr. Nasir,

We’re pleased to inform you that your manuscript has been judged scientifically suitable for publication and will be formally accepted for publication once it meets all outstanding technical requirements.

Kind regards,

Vincenzo Lionetti, M.D., PhD

Academic Editor

PLOS One

Additional Editor Comments (optional):

Reviewers' comments:

Reviewer's Responses to Questions

**Comments to the Author**

1. If the authors have adequately addressed your comments raised in a previous round of review and you feel that this manuscript is now acceptable for publication, you may indicate that here to bypass the “Comments to the Author” section, enter your conflict of interest statement in the “Confidential to Editor” section, and submit your "Accept" recommendation.

Reviewer #1: All comments have been addressed

2. Is the manuscript technically sound, and do the data support the conclusions?

Reviewer #1: Yes

3. Has the statistical analysis been performed appropriately and rigorously? 

Reviewer #1: Yes

4. Have the authors made all data underlying the findings in their manuscript fully available?

Reviewer #1: Yes

5. Is the manuscript presented in an intelligible fashion and written in standard English?

Reviewer #1: Yes

6. Review Comments to the Author

Reviewer #1: Dear Authors,

Thank you for the careful and thorough revision of your manuscript. I have reviewed the second revised version and I am pleased to confirm that all outstanding concerns have been satisfactorily addressed. I have no further requests for revision and am recommending acceptance of your manuscript.

7. PLOS authors have the option to publish the peer review history of their article (what does this mean?). If published, this will include your full peer review and any attached files.

Reviewer #1: **Yes:** Giulio Pellegrini

---

## [Editor Report · Acceptance letter]

PONE-D-25-68283R2

PLOS One

Dear Dr. Nasir,

I'm pleased to inform you that your manuscript has been deemed suitable for publication in PLOS One. Congratulations! Your manuscript is now being handed over to our production team.

Kind regards,

on behalf of

Prof. Vincenzo Lionetti

Academic Editor

PLOS One